# The Effect of Nerolidol Renal Dysfunction following Ischemia–Reperfusion Injury in the Rat

**DOI:** 10.3390/nu15020455

**Published:** 2023-01-15

**Authors:** Fayez T. Hammad, Suhail Al-Salam, Rahaf Ahmad, Javed Yasin, Awwab F. Hammad, Jasmine Abdul Rasheed, Loay Lubbad

**Affiliations:** 1Department of Surgery, College of Medicine & Health Sciences, Al Ain P.O. Box 17666, United Arab Emirates; 2Department of Pathology, College of Medicine & Health Sciences, Al Ain P.O. Box 17666, United Arab Emirates; 3Department of Internal Medicine, College of Medicine & Health Sciences, Al Ain P.O. Box 17666, United Arab Emirates; 4School of Medicine, University of Jordan, Amman 11942, Jordan

**Keywords:** nerolidol, ischemia–reperfusion injury, renal

## Abstract

Efforts to decrease the deleterious effects of renal ischemia–reperfusion injury (IRI) are ongoing. Recently, there has been increasing interest in using natural phytochemical compounds as alternative remedies in several diseases. Nerolidol is a natural product extracted from plants with floral odors and has been proven to be effective for the treatment of some conditions. We investigated the effect of nerolidol in a rat model of renal IRI. Nerolidol was dissolved in a vehicle and administered orally as single daily dose of 200 mg/kg for 5 days prior to IRI and continued for 3 days post IRI. G-Sham (*n* = 10) underwent sham surgery, whereas G-IRI (*n* = 10) and G-IRI/NR (*n* = 10) underwent bilateral warm renal ischemia for 30 min and received the vehicle/nerolidol, respectively. Renal functions and histological changes were assessed before starting the medication, just prior to IRI and 3 days after IRI. Nerolidol significantly attenuated the alterations in serum creatinine and urea, creatinine clearance, urinary albumin and the urinary albumin–creatinine ratio. Nerolidol also significantly attenuated the alterations in markers of kidney injury; proinflammatory, profibrotic and apoptotic cytokines; oxidative stress markers; and histological changes. We conclude that nerolidol has a renoprotective effect on IRI-induced renal dysfunction. These findings might have clinical implications.

## 1. Introduction

Renal ischemia–reperfusion injury (IRI) occurs during many procedures and conditions such as renal transplantation, partial nephrectomy and recovery following systemic hypotension [1]. It leads to several renal alterations, including reduction in the glomerular filtration rate (GFR) [2]. Efforts to reduce the impact of IRI are ongoing, and several agents have been shown to attenuate this effect [3,4,5,6,7,8,9].

Nerolidol (3,7,11-trimethyl-1,6,10-dodecatrien-3-ol), also known as peruviol, is a naturally occurring sesquiterpene alcohol that is present in the essential oils of various plants with floral odors, including *baccharis dracunculifolia* [10,11]. Leaves are the main source of nerolidol, although it also is found in other parts of plants, such as flowers, seeds, fruits, roots and stems. Numerous methods have been used to extract nerolidol from plants, the most common of which is the hydrodistillation method using the Clevenger-type apparatus [12]. Nerolidol was shown to have protective effects under different conditions and in different organs. For instance, nerolidol has a protective effect against myocardial infarction in rats [13] Nerolidol was also found to be neuroprotective against Parkinson’s disease due to its antioxidant properties and lipid peroxidation inhibition activity [14,15]. Moreover, nerolidol exhibits antimicrobial activity against serval microbial agents, such as *Staphylococcus Aureus, Candida Albicans* and *Escherichia Coli* [16]. In the kidney, nerolidol was reported to have a renoprotective effect in lipopolysaccharide-induced acute kidney injury in rats, as it attenuated the rise in blood urea nitrogen (BUN) and creatinine and inhibited an increase in TNF-α and IL-1β [17]. In a thioacetamide-induced kidney injury model, nerolidol was also shown to ameliorate oxidative damage in kidney tissues [18]. However, the effect of nerolidol on renal dysfunction following IRI has not been studied yet; therefore, the aim of this study was to investigate this effect in a rat model of bilateral warm renal IRI.

## 2. Materials and Methods

Studies were performed on male Wistar rats weighing approximately 200 g at the time of IRI. Rats were fed a standard rat chow. Animals were fasted for 12 h before the experimental procedures but had access to water ad libitum. The experimental protocol was approved by the local ethics committee (ERA-2021-8383).

### 2.1. Ischemia–Reperfusion Injury

All surgical procedures were carried out under strict aseptic conditions as previously described [19]. In summary, animals were anesthetized by intraperitoneal injection of pentobarbitone (45 mg/kg). Through a midline abdominal incision, the left and right renal arteries were then exposed and dissected using a surgical microscope. Microsurgical non-traumatic bulldog clamps were applied simultaneously on the left and right renal arteries for 30 min, after which perfusion was restored by releasing the arterial clamps. This was followed by closure of the surgical incision in layers.

### 2.2. Experimental Protocol and Administration of Nerolidol and Vehicle 

Nerolidol (Sigma-Aldrich) was dissolved in 0.5 ml of corn oil as a vehicle and administered by oral gavage immediately after preparation as a single daily dose of 200 mg/kg. The dose was similar to that used in other studies in rats [13,20]. As shown in Figure 1, the treatment started 7 days before the renal IRI surgery and continued for 3 days after the procedure. None of the treated animals showed any adverse effect.

### 2.3. Experimental Groups

The rats were assigned randomly to three groups:

G-Sham (*n* = 10): rats that underwent sham manipulation of both renal arteries and did not receive any medication;

G-IRI (*n* = 10): rats that underwent warm bilateral renal ischemia for 30 min and received only the vehicle (corn oil);

G-IRI/NR (*n* = 10): rats that underwent bilateral renal ischemia for 30 min and received nerolidol dissolved in corn oil.

### 2.4. Sample Collection and Analysis

Urine was collected using metabolic cages for 24 h at 3 different time points: just before the start of nerolidol/vehicle treatment for baseline pretreatment values (Basal), day 6 after nerolidol/vehicle treatment for pre-IRI values (Pre-IRI) and on day 3 after IRI (Post-IRI), and the volume of daily urine at these time points was calculated (Figure 1). Using the tail vein, blood was withdrawn at the same time as urine collection. All samples were frozen at −30 °C for later measurement of urea, albumin and creatinine levels. Seventy-two hours post IRI, the animals were anesthetized using barbiturate (60 mg/kg), and the kidneys were collected and stored in either liquid nitrogen then at −80 °C or in formalin for further assays.

### 2.5. Gene Expression Analysis

A wedge from the middle part of the left kidney containing both the cortex and medulla was excised. It was then snap-frozen in liquid nitrogen and stored at −80 °C for later measurement of gene expression of the following by reverse transcription polymerase chain reaction (RT-PCR):Acute kidney injury markers, i.e., kidney injury molecule-1 (KIM1) and neutrophil gelatinase-associated lipocalin (NGAL);Cytokines involved in the inflammation and fibrosis, i.e., tumor necrosis factor-α (TNFα), transforming growth factor-β (TGF-β1), interleukin-6 (IL-6), interleukin-1 beta (IL-1β) and plasminogen activator inhibitor-1 (PAI-1);The proapoptotic gene p53.Procollagen type-1 (COLA-1); andThe antioxidant enzymes glutathione peroxidase (GPX-1) and glutathione-disulfide reductase (GSR).

Extraction of total RNA from the frozen samples was performed using Qiazol Lysis reagent (Qiagen, Hilden, Germany) as per the manufacturer’s protocol. Estimation of the quantity and quality of the extracted RNA was performed using a NanoDrop 2000 spectrophotometer (Thermo Fisher Scientific, Inc., Wilmington, DE, USA).

Preparation of the strand complementary DNA (cDNA) in duplicates from 1.0 µg of extracted RNA was achieved using a QuantiTect^®^ reverse transcription kit (Qiagen, Hilden, Germany) as per the manufacturer’s protocol. The protocol consisted of genomic DNA removal using the supplied gDNA wipeout buffer, ensuring elimination of interference by the genomic DNA. Subsequently, the prepared cDNA was used as a template for relative gene expression analysis using a TaqMan^®^ hydrolysis probe chemistry kit. The reaction mixture contained 75 ng cDNA, TaqMan universal master mix (Thermo Fisher Scientific Inc., Wilmington, DE, USA), 0.5 µM of forward and reverse primers and 0.25 µM of fluorescent probe (Biosearch Technologies Inc., Petaluma, CA, USA). The probes were FAM-labeled.

Peptidylprolyl Isomerase A (PPIA) housekeeping gene was used for normalization. Its probe was labeled with Quasar 670, enabling multiplexing with the genes of interest. All samples were run in duplicates. At least one primer of all designed PCR primer sets was spanned the exon–exon junction to further exclude any interference of the genomic DNA. Table 1 shows the sequences of primers and probes. The calculated cycle threshold (CT) values were used to estimate the changes in gene expression of target genes using a delta–delta CT formula.

The results are expressed as the mean fold change of gene expression compared to G-Sham. The gene expression of the G-Sham animals was measured, and the average was calculated. This average was given a value of 1. All other groups, including the G-Sham animals were compared to this value.

### 2.6. Enzyme-Linked Immunosorbent Assay (ELISA)

A wedge from the middle part of the left kidney containing the cortex and medulla was excised. The renal tissue concentrations of total glutathione (GSH, Cayman Chemical, Ann Arbor, MI, USA) and malondialdehyde (MDA) (TBARS, Cayman Chemical, Ann Arbor, MI, USA) were measured using ELISA as per the manufacturer’s instructions. The levels were normalized to the total protein concentrations. The concentration was determined by interpolation from a standard curve. Both standards and samples were assayed in duplicate.

### 2.7. Histological Studies

The kidney tissue was washed with ice-cold saline, blotted using filter paper, cassetted and fixed directly in 10% neutral formalin for 24 h. This was followed by dehydration in increasing ethanol concentrations, clearing with xylene and embedding with paraffin. Next, 3 μm sections were prepared from paraffin blocks and stained with hematoxylin and eosin. The stained sections were evaluated blindly using light microscopy.

Microscopic scoring was performed by measuring the percentage of area that showed morphologic changes (tubular dilatation, tubular atrophy, interstitial fibrosis and mononuclear cellular infiltrate) in comparison to the total surface area in each sample. Measurement of the frequency of each histological abnormality was performed using Image J software (NIH, Milwaukee, WI, USA). The degree of abnormality was classified using the following scoring system: score 0, no abnormality; score 1, 1–25%; score 2, 26–50%; score 3, 51–75; score 4, 76–100%.

## 3. Statistical Analysis

Statistical analysis was performed using SPSS V16.0. Results are expressed as mean ± SEM. One-way factorial ANOVA was used for comparison of variables between groups and between different stages (Basal, Pre-IRI and Post-IRI) within each group. A *p* value less than 0.05 was considered statistically significant.

## 4. Results

As demonstrated in Table 2 and Table 3, the basal serum creatinine, serum urea, creatinine clearance, 24 h urinary albumin and albumin/creatinine ratio were similar in all groups (*p* > 0.05 for all variables). Similarly, there was no difference in any of these variables between the Pre-IRI and Basal values in any of the groups (*p* > 0.05 for all variables).

As expected, in the G-Sham, there was no difference in any variable post manipulation compared to the Pre-IRI value (*p* > 0.05 for all variables). In the G-IRI, there was a significant deterioration in renal parameters following IRI (Table 2). For instance, IRI caused an increase in serum creatinine from 0.32 ± 0.05 to 1.51 ± 0.48 mg/dL (*p* < 0.05). In contrast, creatinine clearance decreased to 33.5 ± 7.8 compared to 69.9 ± 6.6 mL/min before IRI (*p* < 0.01). In addition, IRI caused an increase in urinary albumin leakage. The 24 h urinary albumin and albumin/creatinine ratio increased to 0.783 ± 0.111 µg (vs. 0.079 ± 0.009, *p* < 0–001) and 166.1 ± 27.0 (vs. 13.7 ± 2.2, *p* < 0–001), respectively (Table 3).

As shown in Table 2 and Table 3, the administration of nerolidol significantly attenuated IRI-induced alterations in these parameters (*p* < 0.05 for all variables).

### 4.1. Gene Expression Analysis Results

As shown in Figure 2, IRI caused a significant increase in the gene expression of KIM-1 and NGAL, as demonstrated by comparing G-IRI to G-Sham (588.3 ± 92.6 vs. 1.0 ± 0.07 and 24.4 ± 3.5 vs. 1.0 ± 0.08, respectively; *p* < 0.001 for both). Nerolidol administration significantly attenuated this increase in the markers (173.4 ± 51.3 vs. 588.3 ± 92.6, *p* < 0.01 and 13.9 ± 3.1 vs. 24.4 ± 3.5, *p* < 0.05, respectively).

A similar effect was observed in relation to proinflammatory, profibrotic and proapoptotic cytokines (Figure 3, Figure 4 and Figure 5). For instance, IRI led to a significant increase in the gene expression of TGF-β1 in the G-IRI compared to G-Sham (1.88 ± 0.13 vs. 1.01 ± 0.06, *p* < 0.001), whereas nerolidol significantly attenuated this increase (1.46 ± 0.06 vs. 1.88 ± 0.13, *p* < 0.01). Similar effects were observed in the gene expression of TNF-α, PAI-1, IL-6, IL-1β and p53 (*p* < 0.05 for all these mediators).

The gene expression of procollagen type-1 (COLA-1) showed a similar trend, as the IRI caused a significant increase in the gene expression of COLA-1 in G-IRI compared to the G-Sham (3.10 ± 0.41 vs. 1.03 ± 0.12, *p* < 0.001) (Figure 5). Nerolidol significantly attenuated this rise (2.11 ± 0.20 vs. 3.10 ± 0.41, *p* < 0.05).

Nerolidol also significantly attenuated the change in the gene expression of some oxidative stress enzymes, such as GPX-1 and GSR (Figure 6). For instance, IRI led to a significant decrease in the gene expression of GSR (0.62 ± 0.05 vs. 1.02 ± 0.11, *p* < 0.01), and this was attenuated by nerolidol (0.83 ± 0.03 vs. 0.62 ± 0.05, *p* < 0.01).

### 4.2. Enzyme-Linked Immunosorbent Assay (ELISA) Results

As shown in Figure 7, GSH tissue concentration was significantly lower in the G-IRI compared to the G-Sham (3.29 ± 0.12 vs. 5.15 ± 0.23, *p* < 0.001). Nerolidol attenuated this decrease (4.47 ± 0.29 vs. 3.29 ± 0.12, *p* < 0.01). On the other hand, IRI led to a significant increase in MDA tissue concentration (5.94 ± 0.73 vs. 4.09 ± 0.29, *p* < 0.01), and nerolidol resulted in a significant decrease in the MDA concentration (4.76 ± 0.21 vs. 5.94 ± 0.53, *p* < 0.05).

### 4.3. Histological Studies

As shown in Figure 8A,B, the kidneys in G-Sham had normal architecture and histology (score 0). IRI led to diffuse tubular injury and necrosis in 90.2 ± 1.9% (score 4) of the examined tissues (*p* < 0.001 compared to G-Sham). This was associated with the loss of brush border in proximal tubules, tubular dilatation, intratubular necrotic material and secretions and mild mixed inflammatory cell infiltration of the interstitium consisting mainly of lymphocytes (Figure 8C,D). As shown in Figure 8E,F, nerolidol attenuated the effect of IRI, as demonstrated by the fact that only 14.8 ± 4.8% (score 1) of the examined tissues in the G-IRI/NR showed focal cortical tubular injury with tubular dilatation, intratubular necrotic material and secretions and mild mixed inflammatory cell infiltration of the interstitium consisting mainly of lymphocytes (*p* < 0.001 compared to G-IRI).

## 5. Discussion

In the current study, we investigated, for the first time, the effect of nerolidol on renal dysfunction following IRI. We have shown that the administration of this agent prior to and soon after IRI can significantly attenuate IRI-induced alterations in renal functional parameters such as serum creatinine, creatinine clearance and urinary albumin leakage. It also significantly attenuated changes in the gene expression of markers of acute renal injury; proinflammatory, profibrotic and proapoptotic cytokines; and oxidative stress markers. Our data also show a significant improvement in the tissue level of oxidative stress markers and in histopathological changes.

IRI-induced renal damage is caused by both ischemia and reperfusion, leading to hypoxia and excessive production of reactive oxygen species, respectively. This damage is also associated with an inflammatory response characterized by increased production of several cytokines such as proinflammatory and profibrotic cytokines, including TNF-α, TGF-β1 and PAI [21,22,23,24,25,26].

In the current study, nerolidol was shown to have a significant antioxidant effect, as demonstrated by its effects on the gene expression of some oxidative stress enzymes, namely glutathione peroxidase and glutathione-disulfide reductase, as well as on the tissue concentration of glutathione and malondialdehyde. The antioxidant and free radical scavenging properties of nerolidol have been demonstrated in other models, such as memory impairment caused by Trypanosoma evansi in mice [27], thioacetamide-induced oxidative damage in heart and kidney tissue [18] and isoproterenol-induced myocardial damage [13]. The contradictory downregulation in the gene expression of the oxidative enzymes and in their tissue concentration observed in this study has been previously demonstrated by other studies in IRI [28,29] and other conditions [30,31] in which oxidative stress plays a major role in the pathophysiology of the disease process. This downregulation of mRNA has been attributed to the higher turnover of these enzymes at different levels [28]. Regardless of the exact reason, nerolidol led to a significant attenuation of this process in renal IRI.

In addition to its antioxidant properties, the current data demonstrate anti-inflammatory and antifibrotic properties of nerolidol, as shown by the attenuation of the IRI-induced rise in the gene expression of TNF-α, TGF-β1, PAI, IL-1 β and IL-6. TNF-α is a proinflammatory cytokine that is produced by several tissues including renal cells [32]. TGF-β is a profibrotic cytokine that stimulates renal cells to produce extracellular matrix proteins, leading to long-term glomerulosclerosis and tubulointerstitial fibrosis [26,33]. PAI-1 is also a profibrotic cytokine, as it is considered one of the major inhibitors of fibrinolysis [34,35]. All these cytokines are upregulated in the majority of renal diseases [23,24,26], and the effect of nerolidol on these cytokines in renal IRI appears to be similar to its effect in other conditions [17].

The anti-inflammatory and antioxidant properties of nerolidol observed in the current study led to an amelioration of IRI-associated renal functional damage, as shown by the significant attenuation in serum creatinine and creatinine clearance. The improvement in urinary albumin leakage in addition to these parameters indicates that nerolidol not only affects glomerular function but also renal tubular function. This is supported by its significant effects on the expression of some of the markers of acute renal injury, such as NGAL and KIM-1. The latter is strongly expressed and released by injured proximal tubular cells [36], whereas NGAL is synthesized in the thick ascending limb of Henle’s loop and collecting ducts [37]. Therefore, the current data indicate that nerolidol has an effect on different segments of renal tubules, accounting for improvements in renal functions, including urinary albumin leakage.

The current study is subject to some limitations. For instance, owing to the relatively short duration of follow-up after IRI in this study (3 days), it is difficult to assess the long-term histological effects. However, the significant attenuation of the inflammatory and fibrotic cytokines, as well as the attenuation in the gene expression of the proapoptotic gene and procollagen type-1 (COLA-1), indicates that it is likely that the use of nerolidol in IRI could lead to a milder degree of interstitial fibrosis in the long term. Another limitation of this study is the measurement of the gene expression of some cytokines and mediators and the lack of measurement of the protein expression of these factors, such as procollagen-1. Furthermore, in this study, we did not specifically test the tubular functions using electrolyte excretion. However, there are several indicators of an improvement in tubular functions in response to nerolidol administration, such as the improvement in histological features and the attenuation of KIM-1 and NGAL gene expression (vide supra). In addition, we have shown that there was an improvement in urinary albumin leakage under nerolidol treatment. In this regard, some authors believe that albuminuria is primarily caused by proximal tubular damage and impairment in the retrieval and degradation processes [38,39,40]; therefore, the improvement in albuminuria indicates an improvement in renal tubular functions. In addition to the abovementioned limitations, in the current study, we investigated the effect of nerolidol on the kidney but did not test its effect on other organs. Further studies are required to address this point.

The model used in the current study is similar to the clinical scenario of the renal ischemia encountered in conditions such as renal transplantation and partial nephrectomy. These procedures are performed worldwide at an increasing rate; hence, the protective effects of nerolidol demonstrated in the current study might be clinically relevant in such a way that these patients might benefit from taking this agent during the perioperative period. However, further clinical research is required to extrapolate these results to the clinical setup.

In conclusion, the administration of nerolidol before, during and after renal IRI appears to have ameliorated the IRI effects on the renal functional parameters, alterations in markers of kidney injury and histological features. The current data indicate that the protective effects are due to its anti-inflammatory, antifibrotic and antioxidative properties. These findings have potential clinical implications.

## Figures and Tables

**Figure 1 nutrients-15-00455-f001:**
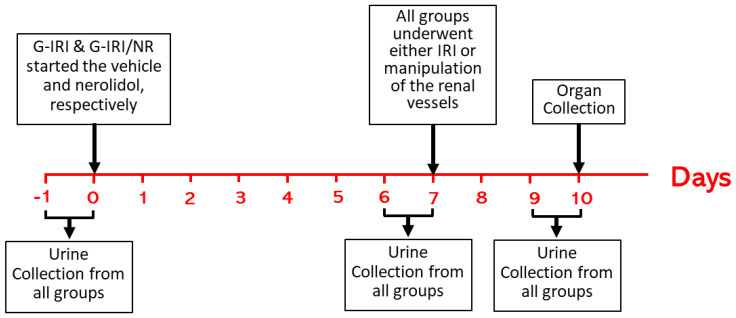
Schematic presentation of the study plan showing interventions in all groups.

**Figure 2 nutrients-15-00455-f002:**
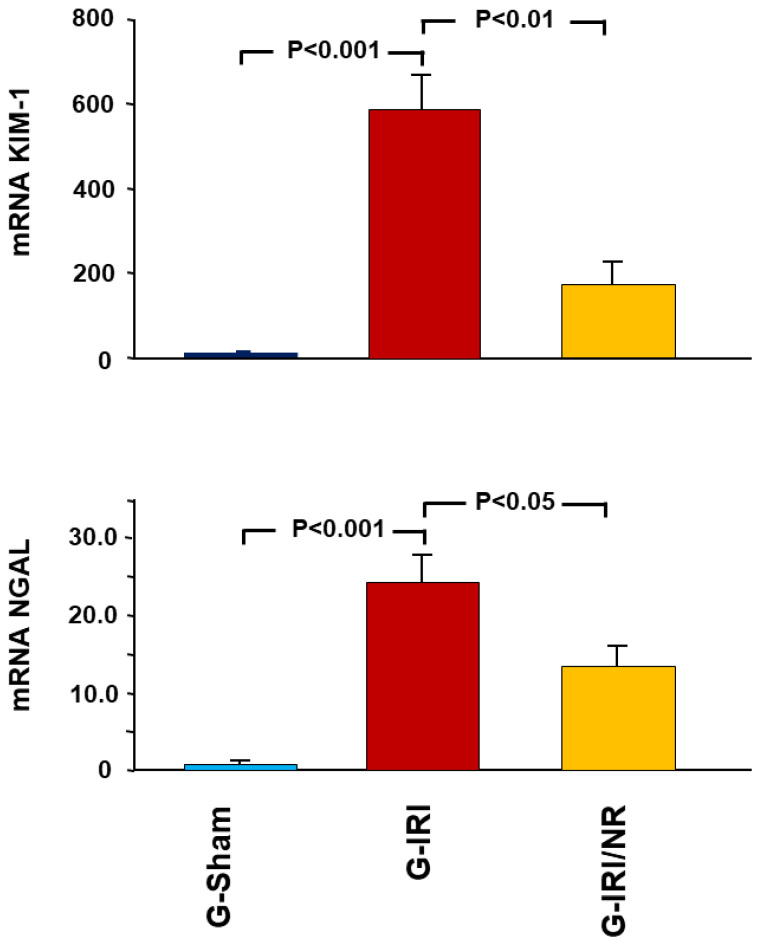
Gene expression of two markers of acute renal injury (KIM-1 and NGAL) in all groups. Values represent mean ± SEM.

**Figure 3 nutrients-15-00455-f003:**
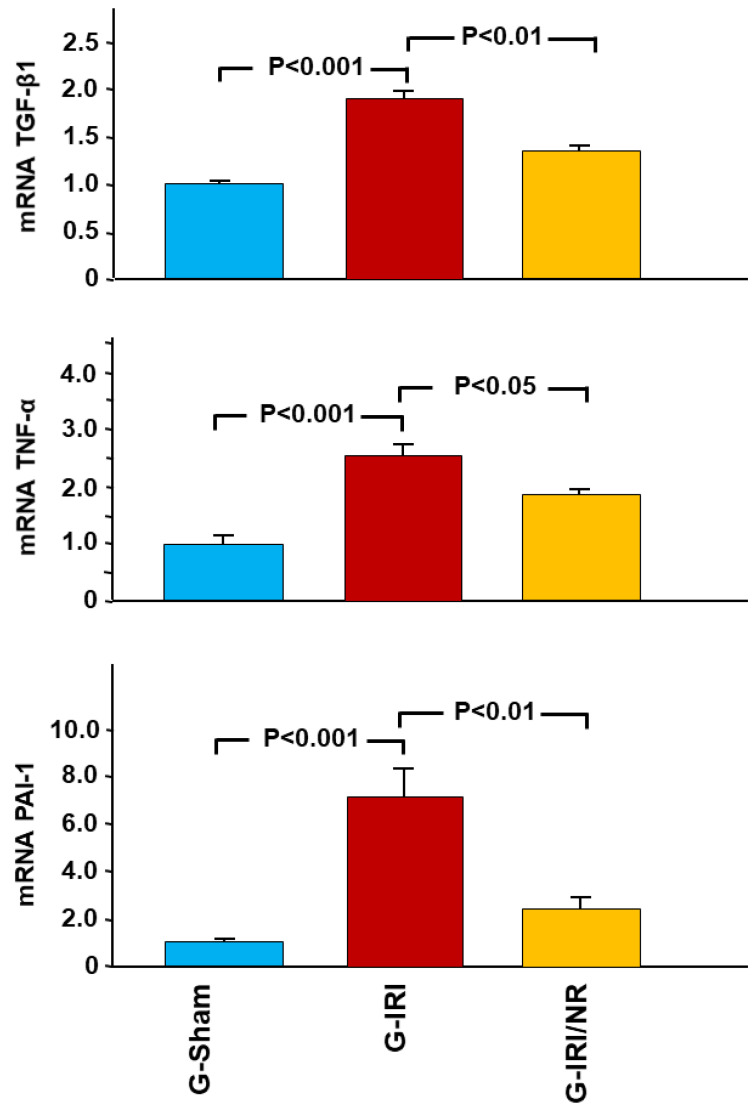
Gene expression of transforming growth factor-β (TGF-β1), tumor necrosis factor-α (TNF-α) and plasminogen activator inhibitor-1 (PAI-1) in all groups. Values represent mean ± SEM.

**Figure 4 nutrients-15-00455-f004:**
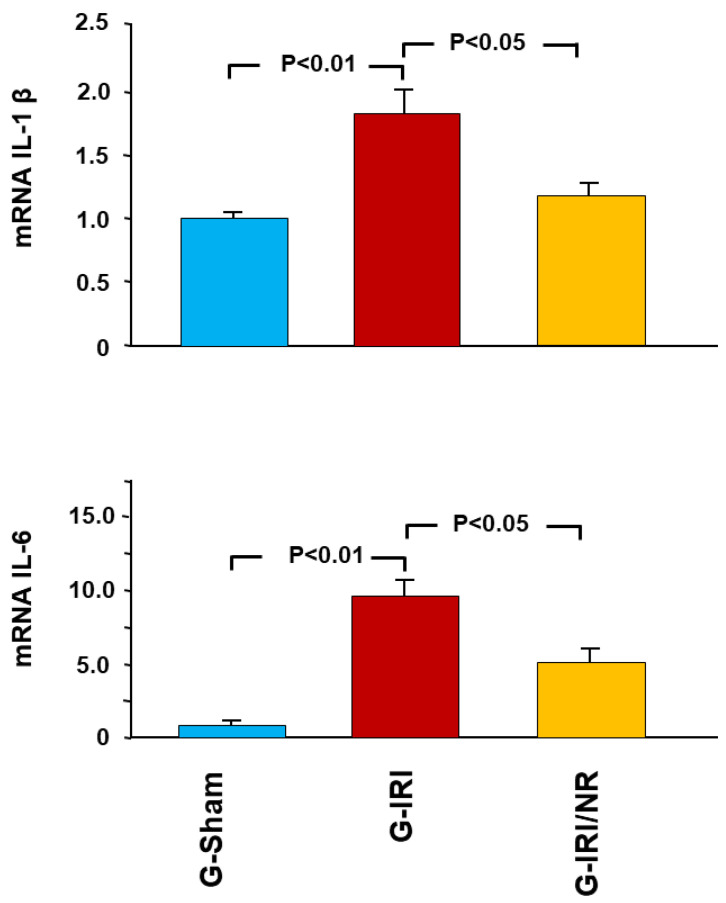
Gene expression of the proinflammatory cytokines interleukin 6 (IL-6) and interleukin 1 beta (IL-1β) in all groups. Values represent mean ± SEM.

**Figure 5 nutrients-15-00455-f005:**
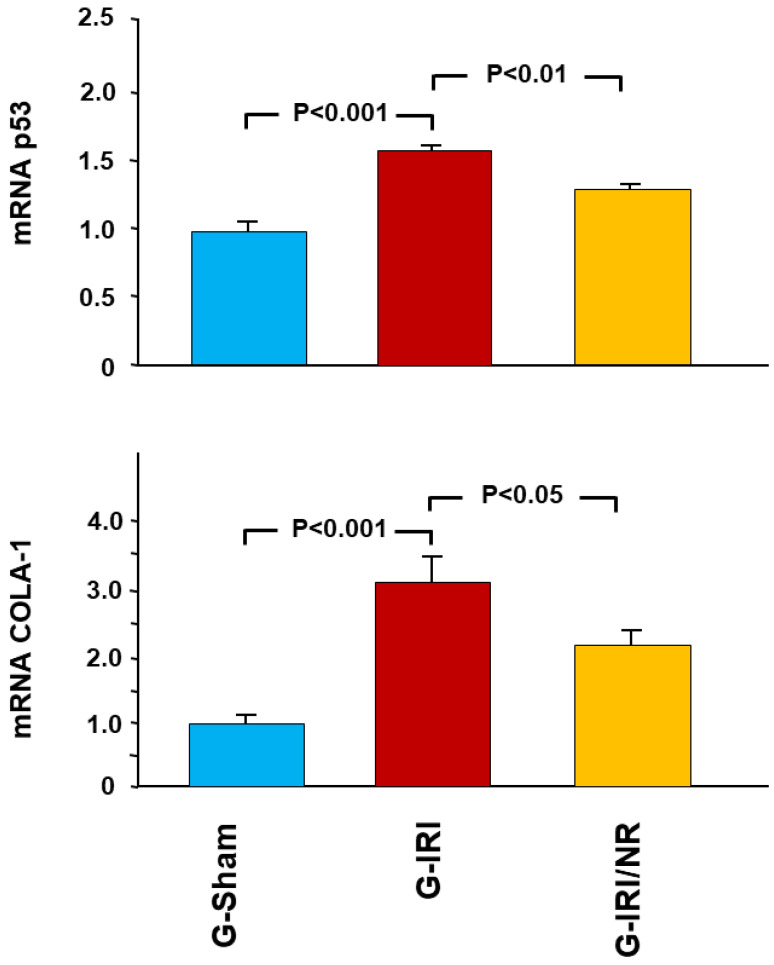
Gene expression of the proapoptotic p53 gene and procollagen-1 (COLA-1) in all groups. Values represent mean ± SEM.

**Figure 6 nutrients-15-00455-f006:**
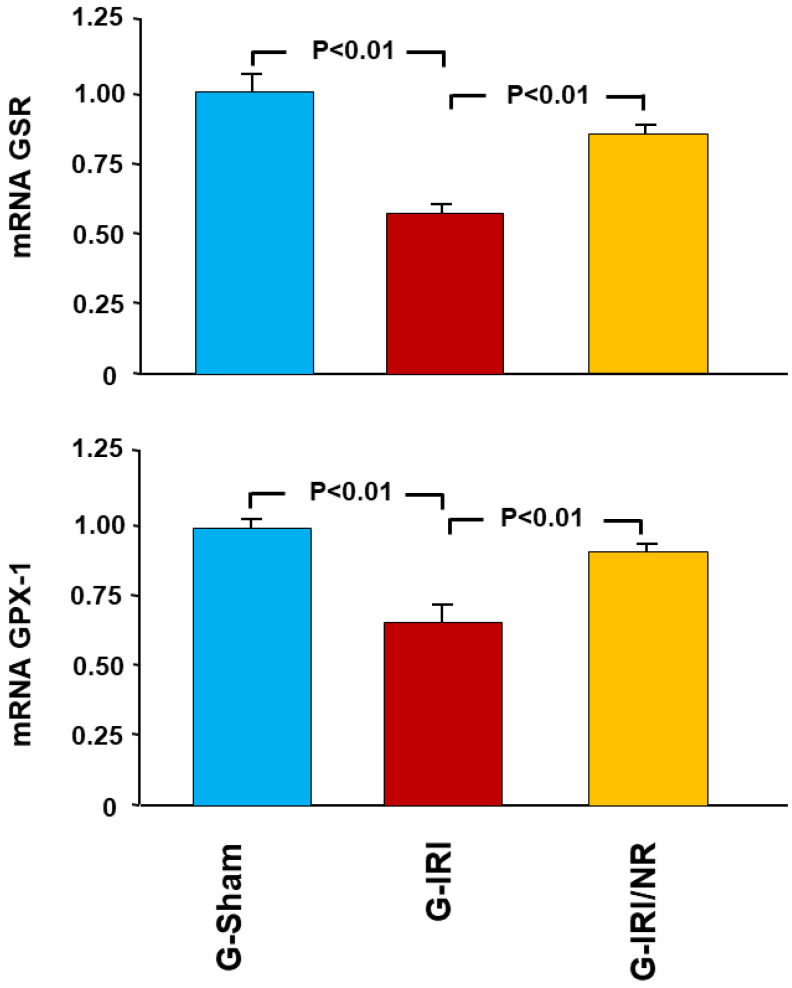
Gene expression of the antioxidant enzymes glutathione peroxidase (GPX-1) and glutathione-disulfide reductase (GSR) in all groups. Values represent mean ± SEM.

**Figure 7 nutrients-15-00455-f007:**
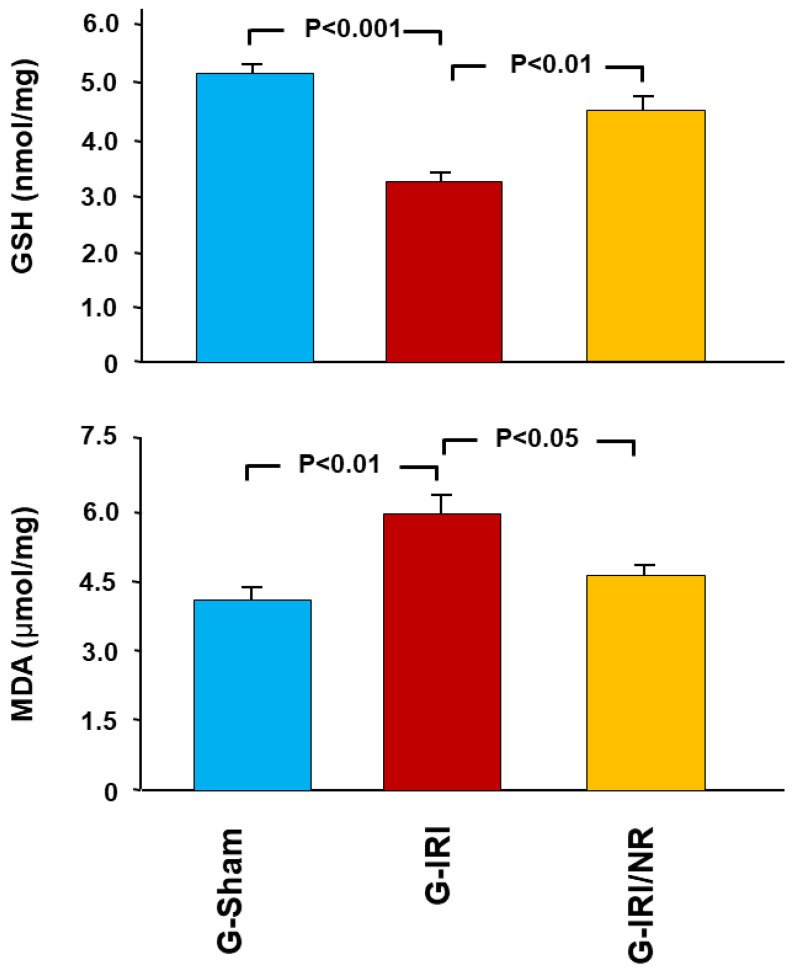
Kidney tissue concentration of gene expression of the glutathione (GSH) and malondialdehyde (MDA, TBARS) measured by ELISA. Values represent mean ± SEM.

**Figure 8 nutrients-15-00455-f008:**
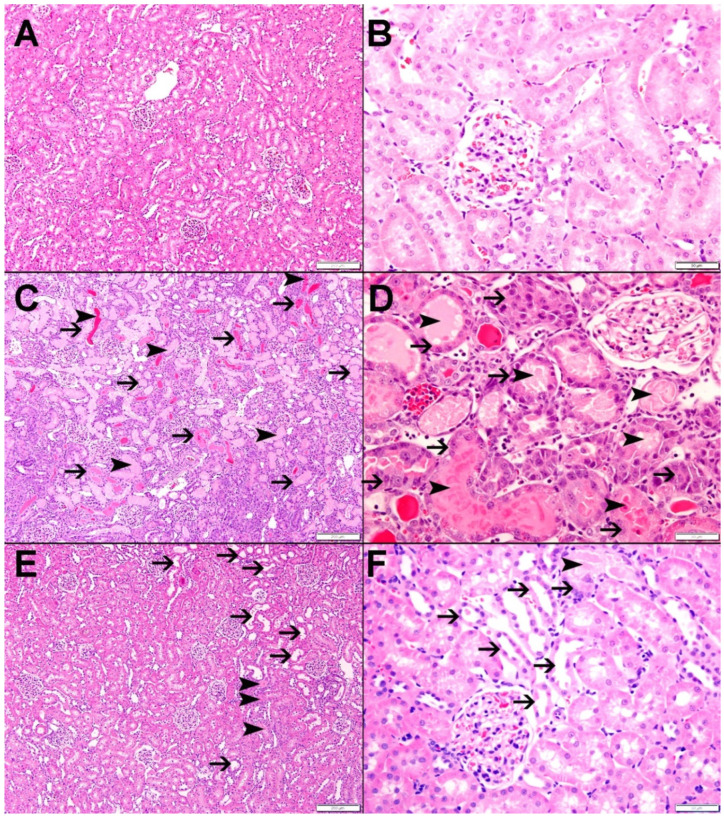
Histological features in all the experimental groups. (**A**,**B**) The kidneys in G-Sham showed normal architecture and histology. (**C**,**D**) Histological features in the G-IRI with diffuse acute tubular necrosis (thin arrow) and intratubular necrotic material and secretions (arrowhead). (**E**,**F**) Histological features in G-IRI/NR with mild acute tubular necrosis and tubular dilation (thin arrow) and intratubular secretion (arrowhead).

**Table 1 nutrients-15-00455-t001:** Forward and reverse primers and fluorogenic probe sequences used for real-time quantitative PCR analysis. KIM-1: kidney injury molecule-1; NGAL: neutrophil gelatinase-associated lipocalin, also called lipocalin 2 (Lcn2); TNF-α: tumor necrosis factor-alpha; TGF-β1: transforming growth factor-β; IL-1β: interleukin 1 beta; IL-6: interleukin 6; TGF-β1: transforming growth factor-β; PAI-1: plasminogen activator inhibitor-1; p53: proapoptotic gene p53; COL1A: procollagen type-1; GSR: glutathione-disulfide reductase; GPx-1: glutathione peroxidase 1; PPIA: peptidylprolyl isomerase A (housekeeping gene).

KIM-1 (NM_173149.2)	Forward	GCCTGGAATAATCACACTGTAAG
	Reverse	GCAACGGACATGCCAACATAG
	Probe	d FAM-TCCCTTTGAGGAAGCCGCAGA-BHQ-1
Lipocalin 2 (LCN2) (NM_130741.1)	Forward	CTGTTCCCACCGACCAATGC
	Reverse	CCACTGCACATCCCAGTCA
	Probe	FAM-TGACAACTGAACAGACGGTGAGCG-BHQ-1
TNF-α (NM_012675.3)	Forward	CTCACACTCAGATCATCTTCTC
	Reverse	CCGCTTGGTGGTTTGCTAC
	Probe	FAM-CTCGAGTGACAAGCCCGTAGCC-BHQ-1
TGF-β1 NM_012620.1	Forward	GTGGCTGAACCAAGGAGACG
	Reverse	CGTGGAGTACATTATCTTTGCTGTC
	Probe	FAM-ACAGGGCTTTCGCTTCAGTGCTC-BHQ-1
PAI-1 (NM_012620.1)	Forward	GGCACAATCCAACAGAGACAA
	Reverse	GGCTTCTCATCCCACTCTCAAG
	Probe	FAM-CCTCTTCATGGGCCAGCTGATGG-BHQ-1
IL-6 (NM_012589.2)	Forward	TCACAGAGGATACCACCCACAACA
	Reverse	CACAAGTCCGGAGAGGAGAC
	Probe	FAM-TCAGAATTGCCATTGCACAACTCT-BHQ-1
IL-1β (NM_031512.2)	Forward	ATGCCTCGTGCTGTCTGACC
	Reverse	GCTCATGGAGAATACCACTTGTTGG
	Probe	FAM-AGCTGAAAGCTCTCCACCTCAATGGA-BHQ-1
p53 (NM_030989.3)	Forward	CGAGATGTTCCGAGAGCTGAATG
	Reverse	GTCTTCGGGTAGCTGGAGTG
	Probe	FAM-CCTTGGAATTAAAGGATGCCCGTGC-BHQ-1
COL1A (NM_053304.1)	Forward	CTGACTGGAAGAGCGGAGAGT
	Reverse	CCTGTCTCCATGTTGCAGTAGAC
	Probe	FAM-ACTGGATCGACCCTAACCAAGGC-BHQ-1
GSR NM_053906.2	Forward	CATCCCTACCGTGGTCTTCAG
	Reverse	ATGGACGGCTTCATCTTCAGT
	Probe	FAM-CCACCCGCCTATCGGGACAGT-BHQ-1
GPx-1 NM_030826.4	Forward	GTGCTGCTCATTGAGAATGTCG
	Reverse	TCATTCTTGCCATTCTCCTGATG
	Probe	FAM-TCCCTCTGAGGCACCACGAC-BHQ-1
PPIA (NM_017101.1)	Forward	GCGTCTGCTTCGAGCTGT
	Reverse	CACCCTGGCACATGAATCC
	Probe	Quasar 670-TGCAGACAAAGTTCCAAAGACAGCA-BHQ-2

**Table 2 nutrients-15-00455-t002:** Serum creatinine, serum urea and creatinine clearance in all groups before administration of nerolidol/vehicle (basal values), after administration and on the third day following IRI of the nerolidol/vehicle, just before ischemia–reperfusion injury (Pre-IRI) (Post-IRI). * Indicates statistical significance when compared to the Pre-IRI within the same group; ^$^ indicates statistical significance compared to the G-IRI group. Values are expressed as mean ± SEM.

		Group		
		G-Sham	G-IRI	G-IRI/NR
S. Creatinine (mg/dL)	Basal	0.31 ± 0.02	0.30 ± 0.02	0.32 ± 0.02
	Pre-IRI	0.29 ± 0.01	0.32 ± 0.05	0.33 ± 0.02
	Post-IRI	0.29 ± 0.02	1.51 ± 0.48 *	0.37 ± 0.02 *^,$^
S. Urea (mg/dL)	Basal	28.9 ± 1.4	29.0 ± 1.4	31.1 ± 2.8
	Pre-IRI	26.6 ± 0.8	30.7 ± 2.8	28.8 ± 2.3
	Post-IRI	25.4 ± 0.6	82.2 ± 19.8 *	31.7 ± 2.1 ^$^
Creatinine Clearance (mL/min)	Basal	62.3 ± 4.3	64.7 ± 9.1	63.5 ± 5.7
	Pre-IRI	72.7 ± 3.4	69.9 ± 6.6	67.0 ± 5.1
	Post-IRI	76.4 ± 3.0	33.5 ± 7.8 *	74.1 ± 2.8 ^$^
	(1)			

**Table 3 nutrients-15-00455-t003:** The 24 h urinary albumin and albumin/creatinine ratio in all groups before administration of the nerolidol/vehicle (basal values), after administration of the nerolidol/vehicle, just before the ischemia–reperfusion injury (Pre-IRI) and on the third day following IRI (Post-IRI). * Indicates statistical significance compared to the Pre-IRI within the same group; ^$^ indicates statistical significance compared to the G-IRI group. Values are expressed as mean ± SEM.

^$^		Group		
*		G-Sham	G-IRI	G-IRI/NR
24-h Urinary Albumin (µg)	Basal	0.073 ± 0.005	0.078 ± 0.013	0.073 ± 0.009
	Pre-IRI	0.077 ± 0.006	0.079 ± 0.009	0.079 ± 0.006
	Post-IRI	0.069 ± 0.005	0.783 ± 0.111 *	0.570 ± 0.078 *^,$^
Albumin/Creatinine Ratio	Basal	13.9 ± 1.1	13.1 ± 2.1	13.6 ± 1.7
	Pre-IRI	13.9 ± 0.9	13.7 ± 2.2	13.8 ± 0.8
	Post-IRI	12.4 ± 0.9	166.1 ± 27.0 *	88.8 ± 11.0 *^,$^

## Data Availability

The data are available on request from the corresponding author.

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
