# Peer review of "The Effect of Nerolidol Renal Dysfunction following Ischemia–Reperfusion Injury in the Rat"

_nutrients, 2023, doi:10.3390/nu15020455_

Round 1

Reviewer 1 Report

The article submitted for review (nutrients-2070025) is a valuable contribution to assessing the effect of natural compounds (nerolidol) on renal dysfunction following renal ischemia-reperfusion injury (IRI)  in a rat model of bilateral warm IRI.

The title of the article reflects the content contained therein. The purpose of the study has been clearly defined. The work structure is appropriate.

References were correctly thematically chosen however, over 30% of them were published before the 2002 year.

My remarks on the reviewed manuscript are as follows:

1)    Page 1 line 44: Expand the information about the natural sources of this substance and the methods of obtaining it.

2)    Page line 2 line 63: Insert the approval document number also here.

3)    Page 4 line 157: Lack of information related to the statistical analysis and tests used to evaluate the results.

4)    Results section: I suggest using 3 levels of statistical significance [p <0.05, p <0.01, p <0.001] for the presented results in the text and the graphs instead of p = ....

5)    Page 4 lines 190-193: The expression, that "the gene expression of procollagen type-1 (COL1A) showed also a similar trend as the IRI caused a significant increase in the gene expression of COL1A compared to the G-191 Sham" is not correct vs. the expression "IRI led to a  significant decrease in the gene expression of GSR compared to G-Sham (0.62±0.05 vs. 187 1.02±0.11, P=0.002) and this was attenuated by the nerolidol (0.83±0.03 vs. 0.62±0.05, 188 P=0.003)" from page 4, lines 186-189. Should be "the gene expression of procollagen type-1 (COL1A) showed an opposite trend as the IRI caused a significant increase in the gene expression of COL1A compared to the G-Sham" or these sentences should be moved one paragraph higher.

6)    Figures 2-7: In the description of the graphs is the information "* and $ indicate statistical significance when compared to the G-Sham and G-IRI, respectively",  but it has not been marked in the graphs in this way.

Author Response

Index of Changes- # nutrients-2070025

The effect of nerolidol on the renal dysfunction following ischemia-reperfusion injury in the rat

The authors would like to thank the Editor and the Reviewers for the positive critical comments which have strengthened the manuscript.

In this document, we have responded to the Reviewers’ comments. The changes refer to the highlighted revised version. In the highlighted version of the manuscript, the underlined text indicates that it has been added whereas strikethrough sign indicates deletion of the text. All modifications are indicated in red. A final neat version was also included.

In addition to addressing the Reviewer’s comments, we made further modifications in the text to render it easier to understand and follow by the reader (Page: 3, Line: 55 and Line:63-65), (Page: 7, Line: 120), (Page: 10, Line: 176, 182 and 188-189), (Page: 11, Line: 189, 205), (Page: 12, Line: 215-216 and 219), (Page: 13, Line: 240 and 246), (Page: 14, Line: 252 and 256), (Page: 15, Line: 280-283) and (Page: 16, Line 306).

Reviewer#1:

The article submitted for review (nutrients-2070025) is a valuable contribution to assessing the effect of natural compounds (nerolidol) on renal dysfunction following renal ischemia-reperfusion injury (IRI) in a rat model of bilateral warm IRI. The title of the article reflects the content contained therein. The purpose of the study has been clearly defined. The work structure is appropriate. References were correctly thematically chosen however, over 30% of them were published before the 2002 year.

My remarks on the reviewed manuscript are as follows:

Comment #1: Page 1 line 44: Expand the information about the natural sources of this substance and the methods of obtaining it

Response: An extra text has been added to address this point (Page: 3, Line: 60-63).

Comment #2: Page line 2 line 63: Insert the approval document number also here.

Response: This has now been added (Page: 4, Line: 80).

Comment #3: Page 4 line 157: Lack of information related to the statistical analysis and tests used to evaluate the results

Response: Somehow, this was dropped during preparing the final draft, apology. This has now been re-added (Page: 8, Line: 166-170).

Comment #4: Results section: I suggest using 3 levels of statistical significance [p <0.05, p <0.01, p <0.001] for the presented results in the text and the graphs instead of p = ....

Response: This has now been changed as per the suggestion in the Results section and in the Figures (Page:10, Line: 180-181, 184, 190 and 192), (Page: 11, Line: 195-197, 201-202, 205-206, 213-214), (Page: 12, Line: 215, 217, 221, 228-229), Figure-2, Figure-3, Figure-4, Figure-5, Figure-6 and Figure-7.

Comment #5: Page 4 lines 190-193: The expression, that "the gene expression of procollagen type-1 (COL1A) showed also a similar trend as the IRI caused a significant increase in the gene expression of COL1A compared to the G-191 Sham" is not correct vs. the expression "IRI led to a  significant decrease in the gene expression of GSR compared to G-Sham (0.62±0.05 vs. 187 1.02±0.11, P=0.002) and this was attenuated by the nerolidol (0.83±0.03 vs. 0.62±0.05, 188 P=0.003)" from page 4, lines 186-189. Should be "the gene expression of procollagen type-1 (COL1A) showed an opposite trend as the IRI caused a significant increase in the gene expression of COL1A compared to the G-Sham" or these sentences should be moved one paragraph higher

Response: The text has been modified to address this point (Page: 11, Line: 199-202, 207-210).

Comment #6: Figures 2-7: In the description of the graphs is the information "* and $ indicate statistical significance when compared to the G-Sham and G-IRI, respectively",  but it has not been marked in the graphs in this way

Response: We checked the Figure Legends and the statistical significances between different groups are indicated by showing the level of significance as numbers as suggested in Comment#4 by Reviewer#1 Figure-2, Figure-3, Figure-4, Figure-5, Figure-6 and Figure-7.

Reviewer 2 Report

Fayez T. Hammad et al. attempt to demonstrate the effect of nerolidol which is natural products and  is extracted from Citrus aurantium, etc. on kidney dysfunction in rats of ischemia-reperfusion injury. As only few studied the effects of nerolidol on kidney dysfunction, their investigation has comparative novelty.  The  subject of this paper is interesting. The effect of nerolidol on kidney dysfunction are demonstrated by their data, however, still there are questions.  See above

1 Is there a rationale for determining the dosage of Nerolidol

2 Did rats treated with Nerolidol show any adverse symptoms? You might want to mention about it.

3  Is the protein expression of Procollagen 1, which is not a liquid substance factor, altered in the same way as mRNA? Please consider additional experiments, such as western blotting and immunostaining.

4 In figure 4, what was the IL-1B gene expression level based on? 

5 In histological studies of the results section, the authors mentioned loss of brush border in proximal tubules.  To me, in figure 8E and 8F, arrows supposedly indicate distal tubules not proximal tubules. If not, they might want to demonstrate that these are proximal tubules with a missing brush border.

6 The conclusion of the paper may be expanded interpretation. What effect did nerolidol have? What mechanism of nerolidol would have improved anti-inflammatory and anti-fibrotic markers? Additional discussion on the details of the mechanism of the nerolidol, or the limitations of these experiments would be needed.

Minor comments

Legends of figure5 and figure 6 are not much these graphs.

Author Response

Index of Changes- # nutrients-2070025

The effect of nerolidol on the renal dysfunction following ischemia-reperfusion injury in the rat

The authors would like to thank the Editor and the Reviewers for the positive critical comments which have strengthened the manuscript.

In this document, we have responded to the Reviewers’ comments. The changes refer to the highlighted revised version. In the highlighted version of the manuscript, the underlined text indicates that it has been added whereas strikethrough sign indicates deletion of the text. All modifications are indicated in red. A final neat version was also included.

In addition to addressing the Reviewer’s comments, we made further modifications in the text to render it easier to understand and follow by the reader (Page: 3, Line: 55 and Line:63-65), (Page: 7, Line: 120), (Page: 10, Line: 176, 182 and 188-189), (Page: 11, Line: 189, 205), (Page: 12, Line: 215-216 and 219), (Page: 13, Line: 240 and 246), (Page: 14, Line: 252 and 256), (Page: 15, Line: 280-283) and (Page: 16, Line 306).

Reviewer #2:

Fayez T. Hammad et al. attempt to demonstrate the effect of nerolidol which is natural products and is extracted from Citrus aurantium, etc. on kidney dysfunction in rats of ischemia-reperfusion injury. As only few studied the effects of nerolidol on kidney dysfunction, their investigation has comparative novelty. The subject of this paper is interesting. The effect of nerolidol on kidney dysfunction are demonstrated by their data, however, still there are questions. See above

Comment #1: Is there a rationale for determining the dosage of Nerolidol

Response: The dose used was similar to the dose used in other studies in rats with protective effect [1, 2]. An extra test was added to address this point (Page: 5, Line: 91-92).

Comment #2: Did rats treated with Nerolidol show any adverse symptoms? You might want to mention about it

Response: Thank you for bringing up this point. None of the treated animals showed any adverse effect. This has now been added to the text (Page: 5, Line: 93-94).

Comment #3: Is the protein expression of Procollagen 1, which is not a liquid substance factor, altered in the same way as mRNA? Please consider additional experiments, such as western blotting and immunostaining

Response: It is hard to ascertain from the current data if the protein expression of procollagen-1 would be altered in the same way as the mRNA, although it is likely. We agree with the Reviewer that further studies are required to test this point. Performing further studies would require further funding which might take some time to obtain and therefore, we believe that this is one of the limitations of the study. An extra text has been added to address this point (Page: 15, Line: 289-291).

Comment #4: In figure 4, what was the IL-1B gene expression level based on? 

Response: The gene expression of IL-1β was measured using reverse polymerase chain reaction as stated in the Methods section (Page: 6, Line: 113-118), (Page: 7, Line: 119-140), (Page: 8, Line: 141-146) and an extra text was added to clarify this point (Page: 6, Line: 116).

Comment #5: In histological studies of the results section, the authors mentioned loss of brush border in proximal tubules.  To me, in figure 8E and 8F, arrows supposedly indicate distal tubules not proximal tubules. If not, they might want to demonstrate that these are proximal tubules with a missing brush border.

Response: Thank you for your valuable comment, the images in Figure-8 E&F was taken from the cortex to demonstrate improvement in acute tubular necrosis following nerolidol treatment. In Figure-8 E&F legend, we did not specify proximal convoluted tubules to be affected as it is difficult to differentiate acutely injured proximal convoluted tubules from distal convoluted tubules because both lack the brush border, however, since the proximal tubules are the predominant cortical tubules to be affected by ischemia reperfusion injury [3], as well as the fact that they constitute the majority of tubules around sampled glomeruli, we have assumed that most of arrowed injured tubules are likely to be proximal tubules, however, the possibility that some of them are distal tubules cannot be ruled out. In addition, distal tubules are likely to be more resistant to IRI than proximal tubules [3]. We have modified the histopathological description accordingly (Page: 12, Line: 226-229).

Comment #6: The conclusion of the paper may be expanded interpretation. What effect did nerolidol have? What mechanism of nerolidol would have improved anti-inflammatory and anti-fibrotic markers? Additional discussion on the details of the mechanism of the nerolidol, or the limitations of these experiments would be needed.

Response: In addition to the already mentioned discussion regarding both the antioxidant, anti-inflammatory and antifibrotic properties of the nerolidol (Page: 13, Line: 245-251), (Page: 14, Line: 253-268), the conclusion has been modified to address this point (Page: 16, Line: 311-316).

Comment #7: Legends of figure5 and figure 6 are not much these graphs

Response: The authors would like to thank the Reviewer for picking up this typo-error. This has now been changed Figure-5 and Figure-6.

Reviewer 3 Report

The study by Hammad et al. demonstrates that Nerolidol, a plant-derived product ameliorates the IRI in male rats. The authors have investigated the effect of Nerolidol in IRI-induced rats related to inflammatory and antioxidant gene expression status. Furthermore, nerolidol treatment improves proteinuria and kidney injury, as evident through histological analysis. Overall, the study provided evidence to support that Nerolidol alleviates the effect caused by IRI in rats. However, the study has many weaknesses and lost enthusiasm due to the following major concerns.

1.       Previously it has been shown that Nerolidol is well known for its beneficial effects in ameliorating kidney injury in LPS and Thioacetamide induced models ((Turk J Pharm Sci. 2022 Feb 28;19(1):1-8; Phytother Res. 2017 Mar;31(3):459-465). The current study didn’t add any new information to the readers other than testing in the IRI model.

2.       Moreover, the authors did not rule out the possibility of the nerolidol effect in other organs, such as the vascular and/or heart. Following IRI, it also interferes with volume expansion and thus increases blood pressure. In this scenario, the study will bring new information to the audience and expand our knowledge on this interesting plant-based compound- Nerolidol.

3.       Furthermore, by assessing the changes in gene expression, concluding it to tubular function may not be sufficient; the authors should determine the tubular function by testing the urinary excretion of electrolytes such as sodium and potassium.

4.       No information was provided regarding statistical analysis, is all the analysis done using one-way ANOVA?

5.       The observed protective phenotype is coming from intervening in inflammation? Oxidative stress? Or renal water-electrolyte balance pathway?

6.       Is there any proposed mechanism for the observed protective phenotype in IRI rats following nerolidol treatment?  

Author Response

Index of Changes- # nutrients-2070025

The effect of nerolidol on the renal dysfunction following ischemia-reperfusion injury in the rat

The authors would like to thank the Editor and the Reviewers for the positive critical comments which have strengthened the manuscript.

In this document, we have responded to the Reviewers’ comments. The changes refer to the highlighted revised version. In the highlighted version of the manuscript, the underlined text indicates that it has been added whereas strikethrough sign indicates deletion of the text. All modifications are indicated in red. A final neat version was also included.

In addition to addressing the Reviewer’s comments, we made further modifications in the text to render it easier to understand and follow by the reader (Page: 3, Line: 55 and Line:63-65), (Page: 7, Line: 120), (Page: 10, Line: 176, 182 and 188-189), (Page: 11, Line: 189, 205), (Page: 12, Line: 215-216 and 219), (Page: 13, Line: 240 and 246), (Page: 14, Line: 252 and 256), (Page: 15, Line: 280-283) and (Page: 16, Line 306).

Reviewer #3:

The study by Hammad et al. demonstrates that Nerolidol, a plant-derived product ameliorates the IRI in male rats. The authors have investigated the effect of Nerolidol in IRI-induced rats related to inflammatory and antioxidant gene expression status. Furthermore, nerolidol treatment improves proteinuria and kidney injury, as evident through histological analysis. Overall, the study provided evidence to support that Nerolidol alleviates the effect caused by IRI in rats. However, the study has many weaknesses and lost enthusiasm due to the following major concerns.

Comment #1: Previously it has been shown that Nerolidol is well known for its beneficial effects in ameliorating kidney injury in LPS and Thioacetamide induced models ((Turk J Pharm Sci. 2022 Feb 28;19(1):1-8; Phytother Res. 2017 Mar;31(3):459-465). The current study didn’t add any new information to the readers other than testing in the IRI model.

Response: We agree with the Reviewer that previous studies had shown protective effects of nerolidol in some renal conditions such as the LPS and thioacetamide-induced renal injury models (Page: 3, Line: 69-73). However, the fact that this substance or any other substance is protective in a condition does not necessarily implicate the same protective effect or magnitude in other conditions due to different pathophysiological changes in each condition. Moreover, renal ischemia reperfusion injury is a much more common condition compared to the conditioned mentioned and much more clinically relevant and the results of this study might have clinical implications. We hope that further clinical studies will lead to the use this agent in patients with renal ischemia reperfusion injury such as those with transplantation, sever hypotension and shock and during renal surgical procedures such as partial nephrectomies (Page: 16, Line: 303-309).

Comment #2: Moreover, the authors did not rule out the possibility of the nerolidol effect in other organs, such as the vascular and/or heart. Following IRI, it also interferes with volume expansion and thus increases blood pressure. In this scenario, the study will bring new information to the audience and expand our knowledge on this interesting plant-based compound- Nerolidol

Response: We totally agree with the Reviewer and indeed, we are currently applying for funding to study these effects in a similar model. An extra text was added to highlight this important point (Page: 16, Line: 300-302).

Comment #3: Furthermore, by assessing the changes in gene expression, concluding it to tubular function may not be sufficient; the authors should determine the tubular function by testing the urinary excretion of electrolytes such as sodium and potassium

Response: We agree with the Reviewer that we did not directly measure the tubular functions by determining the urinary excretion of electrolytes. However, we measured several indicators of tubular functions in response to nerolidol administration, such as the improvement in histological features and the attenuation of the KIM-1 and NGAL gene expression (Page: 14, Line: 272-273) and (Page: 15, Line: 274-279). In addition, we have shown that there was an improvement in the urinary albumin leak with nerolidol. In this regard, some authors believe that albuminuria is primarily caused by proximal tubular damage and impairment in the retrieval and degradation processes [4-6], and hence the improvement in the albuminuria indicates an improvement in renal tubular functions. An extra text has been added to address this point (Page: 15, Line: 292-295) and (Page: 16, Line: 296-300).

Comment #4: No information was provided regarding statistical analysis, is all the analysis done using one-way ANOVA?

Response: Somehow, this was dropped during preparing the final draft, apology. This has now been re-added (Page: 8, Line: 166-170).

Comment #5: The observed protective phenotype is coming from intervening in inflammation? Oxidative stress? Or renal water-electrolyte balance pathway?

Response: From the current data, the protective action was associated with attenuation in both the pro-inflammatory and oxidate stress markers as mentioned in the Discussion section (Page: 13, Line: 245-251), (Page: 14, Line: 252-268), the conclusion has been modified to address this point (Page: 16, Line: 311-316).

Comment #6: Is there any proposed mechanism for the observed protective phenotype in IRI rats following nerolidol treatment?  

Response: As mentioned in Comment#5, from the current data, the protective action was associated with attenuation in both the pro-inflammatory and oxidate stress markers as mentioned in the Discussion section (Page: 13, Line: 245-251), (Page: 14, Line: 252-268), the conclusion has been modified to address this point (Page: 16, Line: 311-316).

References:

  1. Asaikumar, L., et al., Preventive effect of nerolidol on isoproterenol induced myocardial damage in Wistar rats: Evidences from biochemical and histopathological studies. Drug Dev Res, 2019. 80(6): p. 814-823.
  2. Klopell, F.C., et al., Nerolidol, an antiulcer constituent from the essential oil of Baccharis dracunculifolia DC (Asteraceae). Z Naturforsch C J Biosci, 2007. 62(7-8): p. 537-42.
  3. Bonventre, J.V. and L. Yang, Cellular pathophysiology of ischemic acute kidney injury. J Clin Invest, 2011. 121(11): p. 4210-21.
  4. Comper, W.D., et al., Disease-dependent mechanisms of albuminuria. Am J Physiol Renal Physiol, 2008. 295(6): p. F1589-600.
  5. Eppel, G.A., et al., The return of glomerular-filtered albumin to the rat renal vein. Kidney Int, 1999. 55(5): p. 1861-70.
  6. Osicka, T.M., et al., Renal processing of serum proteins in an albumin-deficient environment: an in vivo study of glomerulonephritis in the Nagase analbuminaemic rat. Nephrol Dial Transplant, 2004. 19(2): p. 320-8.

Round 2

Reviewer 2 Report

The authors have rewritten the manuscript comprehensively. I would like to ask  a few more questions.  

1  In figure 4, what was the IL-1B gene expression level based on? What was measured as the standard1?

2 In figure 5, Please remove "." after (COLA-1). 

3 In figure 6, Which spell is correct, GPx or GPX

Author Response

Index of Changes- # nutrients-2070025

The effect of nerolidol on the renal dysfunction following ischemia-reperfusion injury in the rat

The authors would like to thank the Editor and the Reviewers again for the positive critical comments which have strengthened the manuscript.

In this document, we have responded to the Reviewers’ comments. In the highlighted version of the manuscript, revisions made to the manuscript were marked up using the “Track Changes” function. A final neat version was also included.

Reviewer#2:

The authors have rewritten the manuscript comprehensively. I would like to ask a few more questions.  

Comment #1: In figure 4, what was the IL-1B gene expression level based on?What was measured as the standard1?

Response: The authors would like to thank the Reviewer for picking up this graphical error. We have also double-checked all other figures and they are fine. As mentioned in Page: 8, Line: 144-145), the results were expressed as the mean fold change of gene expression compared to the G-Sham. The gene expression of the G-sham animals was measured and the average was calculated. This average was given a value of 1. To this value, all other groups including the animals of the G-Sham were compared. An extra text was added to explain this point (Page: 8, Line: 144-147). We have also amended Figure-6 to show the value of the G-Sham which was not previously properly displayed (Page: 25 and Page: 26). Many thanks again for picking up this error.

Comment #2: In figure 5, Please remove "." after (COLA-1).

Response: This has now been done (Page: 27, Line: 367).

Comment #3: In figure 6, Which spell is correct, GPx or GPX

Response: Thank you for this comment. It is GPX-1 and to be consistent, we have changed in the figure from GPx-1 to GPX-1 (Page: 28, Line: 370) and Figure-6, Page: 29).

Reviewer 3 Report

No further comments.

Author Response

There was no comment from the Reviewer.